# Cilia-enriched oxysterol 7β,27-DHC is required for polycystin ion channel activation

Kodaji Ha[1], Nadine Mundt-Machado [1], Paola Bisignano [2], Aide Pinedo[1], David R. Raleigh [3], Gabriel Loeb [4], Jeremy F. Reiter [5,6], Erhu Cao [7] & Markus Delling [1] ✉

Polycystin-1 (PC-1) and PC-2 form a heteromeric ion channel complex that is abundantly expressed in primary cilia of renal epithelial cells. This complex functions as a non-selective cation channel, and mutations within the polycystin complex cause autosomal dominant polycystic kidney disease (ADPKD). The spatial and temporal regulation of the polycystin complex within the ciliary membrane remains poorly understood. Using both whole-cell and ciliary patch-clamp recordings, we identify a cilia-enriched oxysterol, 7β,27-dihydroxycholesterol (DHC), that serves as a necessary activator of the polycystin complex. We further identify an oxysterol-binding pocket within PC-2 and showed that mutations within this binding pocket disrupt 7β,27-DHC–dependent polycystin activation. Pharmacologic and genetic inhibition of oxysterol synthesis reduces channel activity in primary cilia. In summary, our findings reveal a regulator of the polycystin complex. This oxysterol-binding pocket in PC-2 may provide a specific target for potential ADPKD therapeutics.

Primary cilia are tubular structures that emanate from the surface of most mammalian cells[1]. Cilia house a cilia-specific subset of lipids and proteins that must pass the transition zone at the base of the cilium, which forms a regulated diffusion barrier and provides the structural basis for the cilium as a compartmentalized organelle[2–6]. Very little is known about specific targeting of lipids to the primary cilium. However, several reports suggest that the ciliary membrane comprises a unique set of lipids (such as phosphatidylinositol-4-phosphate, PI(4)P and oxysterols), creating a specialized microenvironment for cilia-specific signal transduction[2,7,8].

Cilia receive and transmit extracellular cues regulating diverse cellular processes ranging from early development to kidney physiology by a subset of cilium-specific signaling complexes[9–11]. Consistent with the essential role of primary cilia as cellular sensors, mutations within genes that encode ciliary proteins underlie a plethora of human diseases (also summarized as ciliopathies) that range from developmental defects to conditions involving brain, motor, and kidney dysfunction[12–16].

Polycystin-1 (PC-1) and PC-2 form a cation-permeant ion channel on the primary cilia of renal epithelial cells[17–19]. PC-1 (encoded by the *PKD1* gene) consists of a large extracellular N-terminal fragment (NTF) and 11 transmembrane domains as the C-terminus fragment (TF)[20]. An intrinsic cleavage mechanism at the G-protein coupled receptor proteolytic site (GPS) cleaves PC-1 into the NTF and CTF, which is indispensable for the function of PC-1[21–23]. PC-2 (encoded by the *PKD2* gene) belongs to the transient receptor potential polycystic (TRPP) ion

[1]Department of Physiology, University of California San Francisco, San Francisco, CA, USA. [2]Department of Molecular Physiology and Biophysics, Vanderbilt University, Nashville, TN, USA. [3]Department of Radiation Oncology, University of California, San Francisco, San Francisco, CA, USA. [4]Department of Medicine, University of California, San Francisco, CA, USA. [5]Department of Biochemistry and Biophysics, Cardiovascular Research Institute, University of California, San Francisco, San Francisco, CA, USA. [6]Chan Zuckerberg Biohub, San Francisco, CA, USA. [7]Department of Biochemistry, University of Utah School of Medicine, Salt Lake City, UT, USA. ✉e-mail: markus.delling@ucsf.edu

channel family and contains six transmembrane domains[14,24,25], providing essential subunits for a functional heteromeric polycystin complex[26–28]. The last six transmembrane domains of PC-1 physically interact with PC-2 to form a heteromeric complex with a ratio of 1:3[29,30]. Mutations in *PKD1* or *PKD2* cause autosomal dominant polycystic kidney disease (ADPKD), which is characterized by continued enlargement of fluid filled cysts within the kidney and other organs[13,17,31–37]. Most ADPKD patients eventually develop end-stage renal disease (ESRD), placing a considerable burden on the patients and healthcare systems[32,38,39]. While human genetics and mouse models of cystic kidney disease both suggest that ADPKD is a ciliopathy, the regulators of polycystin complex function remain poorly understood[9,12]. Strikingly, restoration of PKD protein expression in cystic kidneys of ADPKD animal models reverses cysts, supporting the idea that polycystin activators may be useful for the treatment of ADPKD[40].

Although there is growing appreciation that ciliary membranes have a unique lipid composition, it remains largely unknown how these lipids contribute to ciliary function. For instance, ciliary-specific Hedgehog signaling requires cilia-enriched oxysterols, including 7β,27-dihydroxycholesterol (DHC)[7,8,41,42]. 7β,27-DHC is a cholesterol derivative with two additional hydroxyl groups in the steroid ring and aliphatic chain. 11β-hydroxysteroid dehydrogenase (11β-HSD)[43] is a critical enzyme in oxysterol metabolism. 11β-HSD type 1 (11β-HSD1) catalyzes the stereospecific oxo-reduction of 7κ,27 to 7β,27, while 11β-HSD type 2 (11β-HSD2) mediates the reverse oxidation reaction of 7β,27 to 7κ,27[44,45]. Single-cell RNA-sequencing and proteomics studies have shown that 11β-HSD2 is predominantly expressed along renal tubules, where it essential for mineralocorticoid homeostasis by converting active cortisol to inactive cortisone[46]. However, in contrast to phosphoinositides, our understanding of oxysterols as second messengers is still in its infancy due to a lack of biosensors to visualize oxysterols in vivo. Nevertheless, oxysterols are emerging as a physiologically diverse group of metabolites that may function as second messengers[47–49].

The present study was initiated based on previous observations by our group and others that overexpression of PC-2 homomers and PC-1/PC-2 heteromers in human embryonic kidney (HEK) 293 cells or *Xenopus* oocytes requires gain of function (GOF) mutations in PC-2, such as PC-2$_{F604P}$ or PC-2$_{L677A/N681A}$[28,50–52], to elicit measurable polycystin channels in the plasma membrane. These initial findings were surprising since wild-type (WT) PC-2 subunits form functional channels in the primary cilia of inner medullary collecting duct 3 (IMCD-3) and HEK cells[26–28,53]. Thus, we hypothesized that primary cilia might contain critical cofactors required for polycystin channel activity. Here we compare polycystin channel activity in the plasma and ciliary membranes using whole-cell and ciliary patch-clamp recordings. We find that a cilia-enriched oxysterol, 7β,27-DHC, binds within a cytoplasmic region of PC-2 and activates the polycystin complex. Furthermore, we show that pharmacologic and genetic inhibition of 7β,27-DHC synthesis reduces polycystin channel activity in primary cilia of IMCD-3 cells. This study suggests that oxysterol derivates may be further developed into PC-2 activators that can form the basis of potential ADPKD therapeutics.

## Results

### Cilia-enriched oxysterol 7β,27-DHC activates the polycystin complex on the plasma membrane

We previously had to rely on a GOF mutation (F604P) in PC-2 to characterize basic biophysical properties of the heteromeric polycystin complex in the plasma membrane[28]. Without that GOF mutation, membrane-targeted PC-1/PC-2 complex remained inactive (in agreement with recent reports)[28,51,52], which led us to hypothesize that the plasma membrane compartment is lacking critical cofactors for voltage-dependent activation of the polycystin complex. To test this hypothesis, we developed stable cell lines co-expressing WT PC-2 together with membrane-targeted PC-1 (sPC-1) under a doxycycline-regulated promoter. Surface expression is markedly increased when the endogenous signal peptide is substituted by an Ig k-chain secretion sequence and hemagglutinin (HA) tag[28,51,54](Fig. 1A). Live cell staining with an anti-HA antibody revealed that the heteromeric complex traffics to the plasma membrane of HEK293 cells and ciliary membrane of mouse (m)IMCD-3 cell (Fig. 1B).

After confirming plasma membrane insertion of sPC-1/2, we tested a variety of lipids, previously identified as cilia enriched, for their ability to activate the polycystin complex. We performed whole-cell patch-clamp recordings and applied lipids to either the bath solution or patch pipette solution, which allowed us to selectively apply the lipid to the extracellular or intracellular side of the polycystin complex, respectively (Fig. 1C). Our initial screen of 7 lipids identified 7β,27-DHC as the only potential activator (Supplementary Fig. 1A–C). When included in intracellular solution of the patch pipette, 5 μM 7β,27-DHC, elicited an outward rectifying current from sPC-1/2–expressing HEK293 cells ($139.9 \pm 48.3$ pA/pF, $n = 16$) (Fig. 1D–F). In contrast, non-transfected HEK293 cells did not show any currents above background with 5 μM 7β,27-DHC in the pipette solution ($9.3 \pm 1.3$ pA/pF, $n = 15$), suggesting that 7β,27-DHC specifically activates the polycystin complex (Fig. 1D–F). We next acutely applied 5 μM 7β,27-DHC to the extracellular bath solution of sPC-1/2–expressing HEK cells and performed whole-cell recordings using a ramp pulse. Extracellular application of 7β,27-DHC did not activate sPC-1/2 in HEK293 cells ($10.4 \pm 2.3$ pA/pF, $n = 24$) (Fig. 1G), suggesting that 7β,27-DHC acts via the cytoplasmic leaflet to modulate polycystin activation.

To test whether 7β,27-DHC can also activate either PC-1 or PC-2 homomers in our heterologous expression system, we performed patch-clamp recordings on HEK293 cells expressing either sPC-1 or PC-2 with 5 μM 7β 27-DHC in the patch pipette. Both PC-1– or PC-2–expressing HEK cells failed to generate currents above background (PC-1, $8.3 \pm 2.3$ pA/pF, $n = 12$; PC-2, $10.1 \pm 0.7$ pA/pF, $n = 10$) (Fig. 1D–F). We next tested whether 7β 27-DHC can activate the heteromeric polycystin complex lacking the large extracellular N-terminus. We previously reported that ΔNTF PC-1 (PC-1$^{\Delta NTF}$) effectively trafficked to the plasma membrane when co-expressed with WT PC-2 or PC-2 F604P[28]. However, PC-1$^{\Delta NTF}$/PC-2 did not exhibit any channel activation with intracellular 7β,27-DHC application (Supplementary Fig. 2A and B). We recently also reported that the orthologous PC-1L3/PC-2 heteromer also efficiently traffics to the plasma membrane of HEK293 cells[28]. Still, 7β,27-DHC did not elicit any measurable currents from HEK293 cells expressing sPC-1L3/PC-2 (Supplementary Fig. 2A and B). Collectively, our data suggest that the heteromeric PC-1/PC-2 polycystin complex requires both extracellular N-terminus and intracellular 7β,27-DHC to form a functional channel in the plasma membrane.

We next tested the permeation of larger synthetic cations, such as N-methyl-d-glucamine (NMDG (+)). Previous electrophysiological recordings from primary cilia enriched in either PC-2 F604P or PC-2 WT suggest that NMDG impairs ion permeation and thus reduces outward currents. We observed that both sPC-1/2–dependent outward and inward currents decreased after the charge carrier was switched from Na$^+$ ($60.77 \pm 23.02$ pA/pF at +100 mV; $-9.29 \pm 3.43$ pA/pF at $-100$ mV) to NMDG$^+$ ($+13.94 \pm 4.05$ pA/pF at +100 mV; $-2.80 \pm 1.37$ pA/pF), in agreement with previous reports[25,27,55]($n = 5$) (Fig. 1H, I). As expected, intracellular solutions with NMDG as charge carrier completely abolished ion permeation (Supplementary Fig 2C and D).

### Higher concentrations of 7β,27-DHC increase open probability at negative potentials

We next hypothesized that 7β,27-DHC may interact directly with the polycystin complex. We measured polycystin activation in the inside-out single-channel configuration of sPC-1/2–overexpressing HEK293

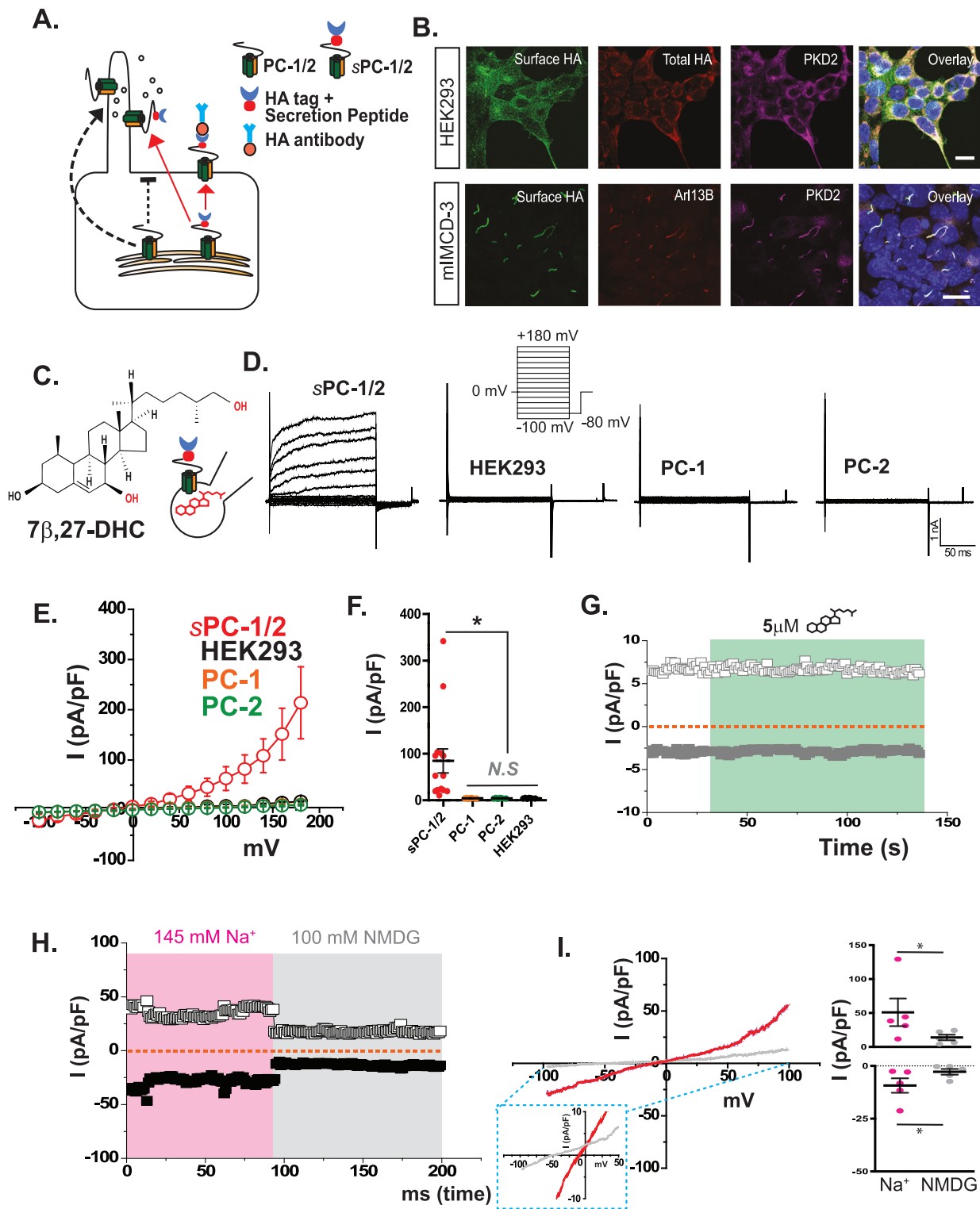

cell membranes and applied 1 μM, 2.5 μM, 5 μM, and 50 μM 7β,27-DHC to the bath solution (Fig. 2A). In this configuration, the cytoplasmic leaflet is exposed to the bath solution. In single-channel recordings, 7β,27-DHC activated sPC-1/2 in a concentration-dependent manner (G: 1 μM 57.2 ± 1.9 pS, $n = 3$; G: 2.5 μM 66.4 ± 1.0 pS $n = 3$; G: 5 μM 63.75 ± 9.65 pS $n = 4$; G: 50 μM 76.87 ± 4.28 pS $n = 4$; Fig. 2B, Supplementary Fig. 3A–D). Notably, 50 μM 7β,27-DHC also significantly increased open probability at −100 mV membrane potential (Fig. 2C). Thus, we conclude that 7β,27-DHC behaves as a gating modifier to promote the open state of the polycystin complex.

## 7β,27-DHC intercalates within PC-2 subunits

Although 7β,27-DHC activates heterologously expressed sPC-1/2, so far it remains unclear whether activation is mediated by a direct lipid-channel interaction. We determined specificity by testing structural homologs of 7β,27-DHC, such as 7β-hydroxycholesterol (7β-HC), 7α-HC, and 7α,27-DHC in their ability to activate the polycystin complex (Fig. 3A).Interestingly, 7β-HC, 7α-HC, and 7α,27-DHC all failed to generate significant outward rectifying currents compared to 7β,27-DHC (Fig. 3B), suggesting that the activation is enantioselective and pointing towards a specific binding pocket within the polycystin complex.

**Fig. 1 | 7β,27-DHC activates the polycystin complex on the plasma membrane.**
**A** Strategy to express the polycystin complex in the plasma membrane. The black dotted arrow indicates ciliary trafficking of PC-1 with endogenous signal peptide and PC-2. Red arrow indicates redirected plasma membrane trafficking using a IgG kappa-derived secretion peptide on PC-1[28]. **B** Immunofluorescent staining of HEK293 and IMCD-3 cells overexpressing the polycystin complex. Surface HA staining indicates that polycystin complex accumulates in plasma membrane of HEK cells (Top) or cilia (IMCD-3 cells, bottom). Green, surface HA; Red, total HA or cilia marker Arl13b; Magenta, PC-2; Images are representative for at least three independent experiments. scale bar = 10 μM. **C** Structure of 7β,27-dihydrocholesterol (DHC). **D** Whole-cell patch-clamp traces recorded from parental HEK293 cells or overexpressing sPC-1/2, PC-1 or PC-2. Voltage step pulse from −100 mV to +180 mV in +20 mV increments with 0 mV holding potential and 5 μM 7β,27-DHC in the pipette solution. **E, F** Current (I)−voltage (V) relationship and current density for HEK293 cells overexpressing sPC-1/2, PC-1 or PC-2. Each value

was obtained at the maximum current (pA) from −100 mV to +180 mV and divided by capacitance (pF); sPC-1/2 (red, $n = 16$), PC-1 (orange, $n = 12$), PC-2 (green, $n = 10$), and HEK293 ($n = 15$). **F** Quantification of current density I (pA/pF) at +180 mV. Two-tailed unpaired Student's t-test *$P < 0.05$. **G** Effect of extracellular application of 7β,27-DHC on polycystin channel activation. 5 μM 7β,27-DHC was applied in bath solution. Ramp pulse applied from −100 mV to +100 mV with 500 ms duration. Each value (pA) at −100 mV (gray square) and +100 mV (gray blank square) is divided by capacitance (pF) and plotted over time. Orange line indicates holding potential (0 mV). **H, I** Extracellular NMDG inhibits 7β,27-DHC evoked PC−1/2 currents. Blank and black squares indicate the current density (pA/pF) obtained at +100 mV and −100 mV, respectively. **i** Left. I−V relationship of polycystin currents in presence of Na⁺ (red) and NMDG (gray). Blue insert shows reversal potential shift as the extracellular solution changes from Na⁺ to NMDG⁺. Right. (**E**) I (pA/pF) at +100 mV and −100 mV before (red) and after (gray) NMDG perfusion. $n = 5$. Two-tailed paired Student's t-test, *$P < 0.05$. Error bars: mean ± SEM.

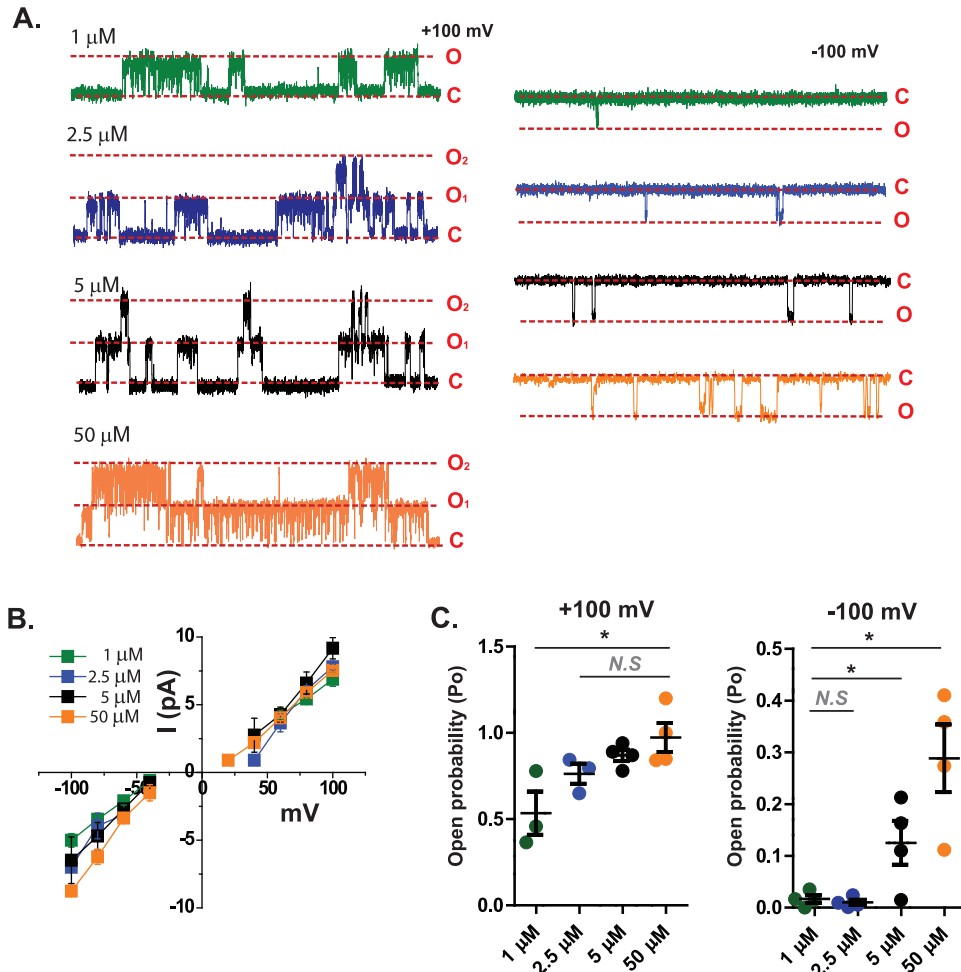

**Fig. 2 | Higher concentration of 7β,27-DHC treatment increase open probability of sPC-1/PC-2 inward currents. A** Representative inside-out single channel recordings of cells overexpressing sPC-1/2 with intracellular application of 1 μM to 50 μM 7β,27-DHC. Red dotted lines indicate closed (C) and open (O) states of a single channel. **B** Single-channel conductance after intracellular application of 1 μM to 50 μM 7β,27-DHC. green: 1 μM ($n = 3$), blue: 2.5 μM ($n = 3$), black: 5 μM ($n = 4$) and

orange: 50 μM ($n = 4$) **C** Normalized absolute open probability of sPC-1/2 obtained from −100 mV to +100 mV during 10 s of recording. The absolute open probability was normalized to the maximum open probability. 1 μM ($n = 3$), 2.5 μM ($n = 3$), 5 μM ($n = 4$) 50 μM ($n = 4$). Two-tailed unpaired Student's t-test, *$P < 0.05$. Error bars: mean ± SEM.

Interestingly, two recent studies suggested that PC-2 is a sterol-binding protein[56,57], suggesting that 7β,27-DHC directly interacts with PC-2. To test this hypothesis, we performed molecular dynamics simulations using 7β,27-DHC with the published PC-2 structure[25]. We focused on the cytoplasmic domains of the PC-2 complex based on our prior finding (Fig. 1G). As shown in Fig. 3C and D, molecular dynamics

simulations identified a putative oxysterol-binding pocket formed by the pre-S1 helix and the S4-S5 linker (Fig. 3d). Specifically, E208 within pre-S1 and R581 in the S4-S5 loop were predicted to interact with 7β,27-DHC (Fig. 3D). In support of these simulations, a previous study predicted the same region as a binding pocket for phosphoinositide[57]. To test whether phosphoinositides and 7β,27-DHC may compete for the

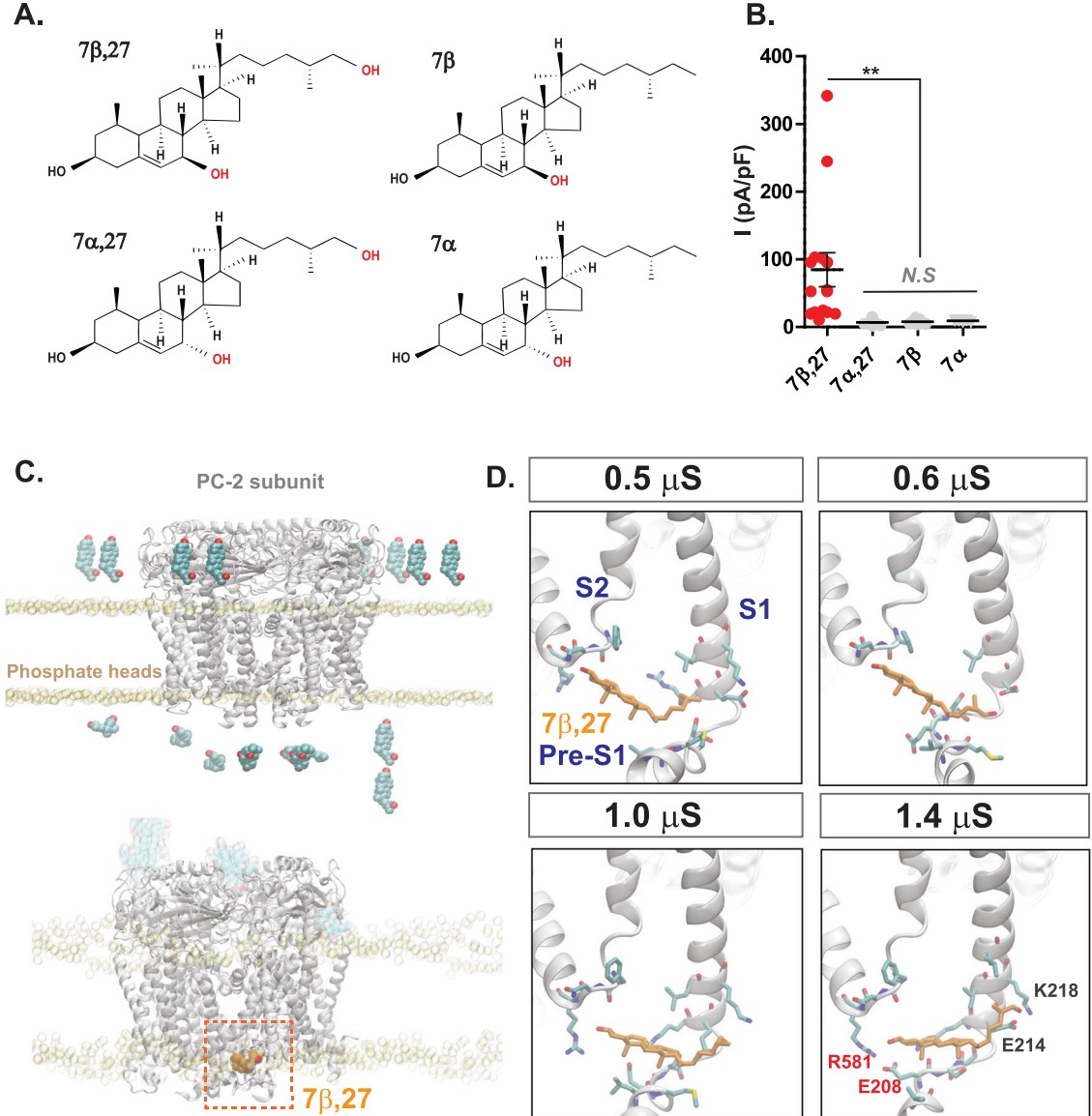

**Fig. 3 | 7β,27-DHC binds to PC-2 subunit. A** Other oxysterols closely related 7β,27-DHC. Molecular structures of three oxysterols, 7β–HC, 7α,27-DHC, and 7α– HC closely related to 7β,27-DHC. Hydroxyl groups colored in red indicate unique hallmarks in each structure. **B** Current density from whole-cell recordings expressing sPC-1/2. 5 μM 7β, 7α,27-DHC, and 7α were included in the intracellular recording solution. Current density is plotted at +180 mV. 7β,27 ($n = 16$), 7α,27 ($n = 13$) 7β ($n = 7$) 7α ($n = 10$), Two-tailed unpaired Student's $t$-test, **$P < 0.01$. Data are presented as mean values ± SEM. **C** Molecular dynamics simulations of oxysterol binding within the PC-2 structure (PDB: 5T4D). The top image shows the system set-up with eight 7β,27-DHC molecules placed in solution (cyan spheres), above the upper leaflet and below the lower leaflet. 7β,27-DHC molecules are free to bind to either extracellular or intracellular leaflet of PC-2 during simulation. Bottom image represents a binding event of one 7β,27-DHC molecule (orange spheres) to the intracellular leaflet of PC-2. Protein is represented as cartoon, and phosphate heads of lipids as yellow spheres to show membrane boundaries. **D** Snapshots of 7β,27-DHC binding mode during molecular dynamics simulations. Once one 7β,27-DHC molecule binds to the intracellular region between pre-S1 and the S4-S5 linker, it stably interacts with E208 and R581 throughout 1.4 μs of simulation time. The protein is shown as white cartoon, 7β,27-DHC as orange/red sticks and residues within 4 Å as cyan/red/blue sticks.

same binding pocket, we added 5 μM PI(3,5)P₂, PI(4,5)P₂, and PI(4)P together with 5 μM 7β,27-DHC to the intracellular solution (Supplementary Fig. 1A and B). Remarkably, 5 μM PI(3,5)P₂, PI(4,5)P₂ decreased the current density of sPC-1/2 treated with 7β,27-DHC, while only PI(4)P completely abolished the channel activation by oxysterol, suggesting high affinity competition of ciliary enriched PI(4)P with 7b,27-DHC (Supplementary Fig. 1A and B).

To test the importance of E208 and R581 for 7β,27-DHC−dependent channel activation, we mutated both amino acids to alanine (hereafter called sPC-1/2$_{E208A}$ and sPC-1/2$_{R581A}$). While sPC-1/2$_{E208A}$ and sPC-1/2$_{R581A}$ still localized to the plasma membrane (Fig. 4A, B), both mutants failed to generate currents above background in the presence of 5 μM 7β,27-DHC (sPC-1/2$_{E208A}$ 8.7 ± 0.8 pA/pF, $n = 10$ and sPC-1/2$_{R581A}$ 6.8 ± 0.9 pA/pF, $n = 12$, Fig. 4C top and D). These data strongly suggest that E208 and R581 are critical for 7β,27-DHC−mediated activation of the polycystin complex. Interestingly, Zheng et al. also identified PC-2 R581 as a critical amino acid for channel activation[55]. To confirm that the effect of mutation of E208 and R581 is specific to 7β,27-DHC−mediated polycystin activation and does not impair general channel function, we introduced E208A or R581A together with the GOF mutation F604P (sPC-1/2$_{E208A-F604P}$ and sPC-1/2$_{R581A-F604P}$; Fig. 4C, bottom and D). Both double mutants generated a smaller, but still measurable outwardly rectifying current, sPC-1/2$_{E208A-F604P}$ (33.0 ± 10.0 pA/pF, $n = 6$) and sPC-1/2$_{R581A-F604P}$

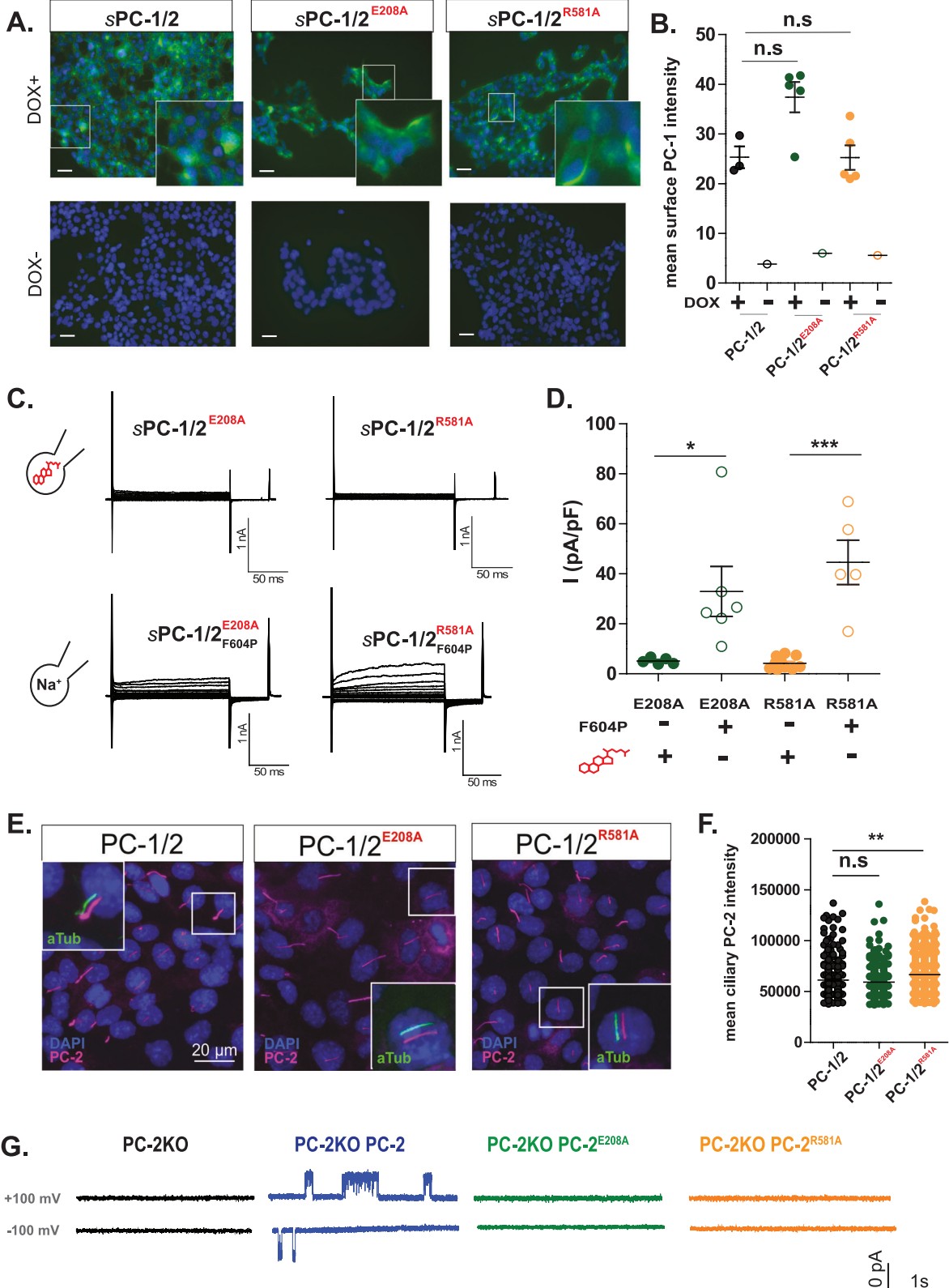

(44.6 ± 8.9 pA/pF, $n = 5$), suggesting that E208 and R581 are specifically required for 7β,27-DHC–dependent activation.

To test whether sPC-1/2$_{E208A}$ and sPC-1/2$_{R581A}$ can still elicit ciliary currents, we performed excised ciliary single-channel recordings from IMCD-3 cells with ablated endogenous PC-2 expression and re-expressed WT PC-2, PC-2$_{E208A}$ and PC-2$_{R581A}$ (Fig. 4G, PC-2 $n = 4$; PC-2$_{E208A}$ $n = 21$; PC-2$_{R581A}$ $n = 22$)[26,28]. While both mutants still localized on the ciliary membrane (Fig. 4E, F), only cells with re-expressed WT PC-2 showed basal activity in the ciliary membrane, while both E208 and R581 mutants remained silent. These results further support the idea that E208 and R581 are critical within the 7β,27-DHC binding pocket.

**Fig. 4 | E208 and R581 within PC-2 are key amino acids required for 7β,27-DHC-dependent activation. A**, **B** Mutation of E208A or R581A does not affect plasma membrane or ciliary trafficking of the polycystin complex. Live cell immunostaining of HEK cells (top: DOX-induced, bottom: DOX-uninduced) stably expressing sPC-1/PC-2, sPC-1/PC-2$_{E208A}$ or sPC-1/PC-2$_{R581A}$ using an anti-HA antibody (green), indicative of sPC-1 trafficking. Data represent one intensity value per image. sPC-1/PC-2 (+Dox: $n = 3$, −Dox: $n = 1$), sPC-1/PC-2$_{E208A}$ (+Dox: $n = 5$, −Dox: $n = 1$), sPC-1/PC-2$_{R581A}$ (+Dox: $n = 5$, −Dox: $n = 1$). Two-tailed unpaired student's T-test. **C** sPC-1/PC-2 E208A or R581A are insensitive to 7β,27-DHC-dependent activation. Whole-cell patch-clamp recordings of HEK cells stably expressing sPC-1/PC-2, sPC-1/PC-2$_{E208A}$ or sPC-1/PC-2$_{R581A}$ and respective double mutant with additional GOF F604P. The same voltage step pulse protocol from Fig. 1D was applied to test binding mutants. sPC-1/2$_{E208A}$ and sPC-1/2$_{R581A}$ were tested with 5 μM 7β,27-DHC added to the intracellular solution, while the binding mutants with F604P (sPC-1/2$_{2E208A-F604P}$ and sPC-1/2$_{R581A-F604P}$) were tested without 5 μM 7β,27-DHC. **D** Comparison of current density at +180 mV from whole-cell patch-clamp recordings of the two oxysterol-binding mutants. The current amplitudes at +180 mV with E208A (green, $n = 10$) and R581A (yellow, $n = 11$) were compared to those of the double mutants, E208A-F604P (green, $n = 6$) and R581A-F604P (yellow, $n = 5$). Two-tailed unpaired Student's $t$-test *$P < 0.05$, ***$P < 0.0001$. **E**, **F** Immunofluorescent imaging analysis of ciliary expression of sPC-1 with PC-2$_{E208A}$ or PC-2$_{R581A}$. Ciliary localization of sPC-1 co-expressed with PC-2, PC-2$_{E208A}$ and PC-2$_{R581A}$ was confirmed using anti-PC-2 (red) and anti-acetylated tubulin antibodies (cilia marker, green). Each n indicates the number of cilia for each group. PC-2 ($n = 328$), PC-2$_{E208A}$ ($n = 289$) and PC-2$_{R581A}$ ($n = 374$). **$P < 0.005$ between the indicated groups tested by One-way ANOVA. **G** Representative excised ciliary single channel recordings. Single channel recordings of parent PC-2 knockout IMCD-3 cells or overexpressing sPC-1 with PC-2, PC-2$_{E208A}$ or PC-2$_{R581A}$ without intracellular 7β,27-DHC application are shown at +100 mV and −100 mV. Error bar, mean ± SEM.

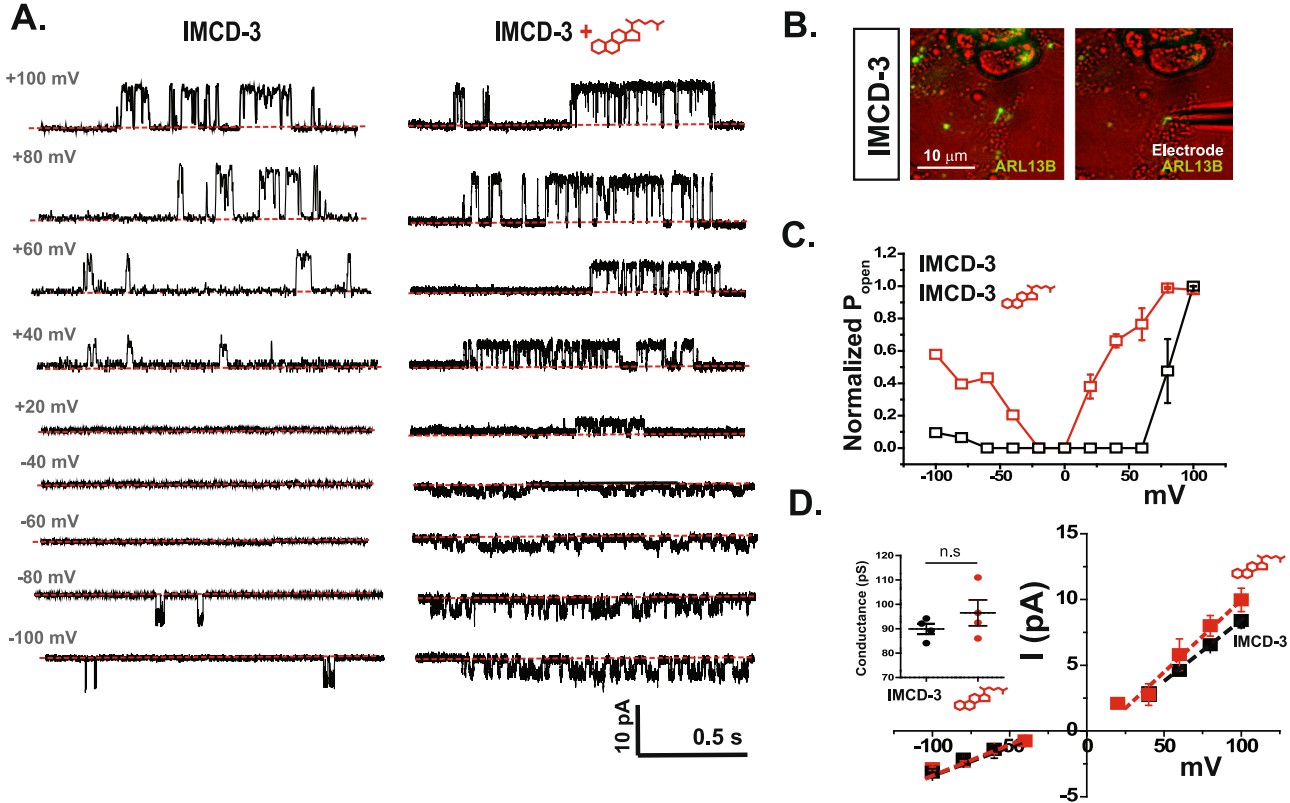

**Fig. 5 | 7β,27-DHC potentiates ciliary polycystin-mediated currents.**
**A** Representative excised inside-out single channel recordings of primary cilia in IMCD-3 cell without (left) and with 5 μM 7β,27-DHC (right). Single channel recordings were obtained between +100 mV and −100 mV at 20 mV increments. Red dotted line indicates the close state of the channel. **B** Representative image of ARL13B-GFP expressing IMCD-3 cells. Primary cilia (green) form a giga-ohm seal with glass electrode (right). **C** Normalized absolute open probability of ciliary channels. P$_{open}$ of ciliary channels from IMCD-3 cells without ($n = 4$) and with 5 μM 7β,27-DHC ($n = 4$) **D** Single-channel conductance of ciliary channel recordings shown in 3 Å. Black and red squares indicate the averaged current amplitudes of ciliary channels from IMCD-3 cells without ($n = 4$) or with 5 μM 7β,27-DHC ($n = 4$) application, respectively. Dotted line indicates the fitting to the linear equation. Bar graph insert shows comparison of ciliary channel conductance without or with 5 μM 7β,27-DHC. Error bars: mean ± SEM.

## 7β,27-DHC application to the ciliary compartment further potentiates ciliary polycystin channels

We next asked whether exogenous application of 7β,27-DHC to the inner leaflet of the ciliary membrane can further potentiate ciliary polycystin channels. We performed excised ciliary inside-out patch-clamp recordings (Fig. 5B) to measure the activation of endogenous polycystin channels by 7β,27-DHC using ARL13B-EGFP−expressing mouse (m)IMCD-3 cells. We found that in the presence of 7β,27-DHC, ciliary polycystin channels started to open at holding potentials of +40 mV and −20 mV, whereas they remained closed at these potentials without the exogenous addition of 7β,27-DHC. These data suggest that exogenously applied 7β,27-DHC can further potentiate the ciliary polycystin complex (Fig. 5A). Furthermore, the absolute open probability of the ciliary polycystin channel with 7β,27-DHC was right-shifted at a negative potential, indicating channel activation in the physiological range (Fig. 5C). The conductance of ciliary polycystin channels did not change with application of 7β,27-DHC (G$_{cilia+7β,27-DHC}$: 96.5 ± 5.3 pS, $n = 4$; G$_{cilia}$: 89.9 ± 2.2 pS, $n = 4$), indicating that 7β,27-DHC is an allosteric polycystin modulator and does not affect ion permeation (Fig. 5D). Next, we determined sensitivity of sPC-1/2 F604P to 5 μM 7β,27-DHC. R581 and F604 are in proximity, raising the interesting possibility that alanine to proline mutation at position 604 could shunt oxysterol regulation of the polycystin complex. Interestingly, 7β,27-DHC did not further potentiate PC-2 F604P, thereby

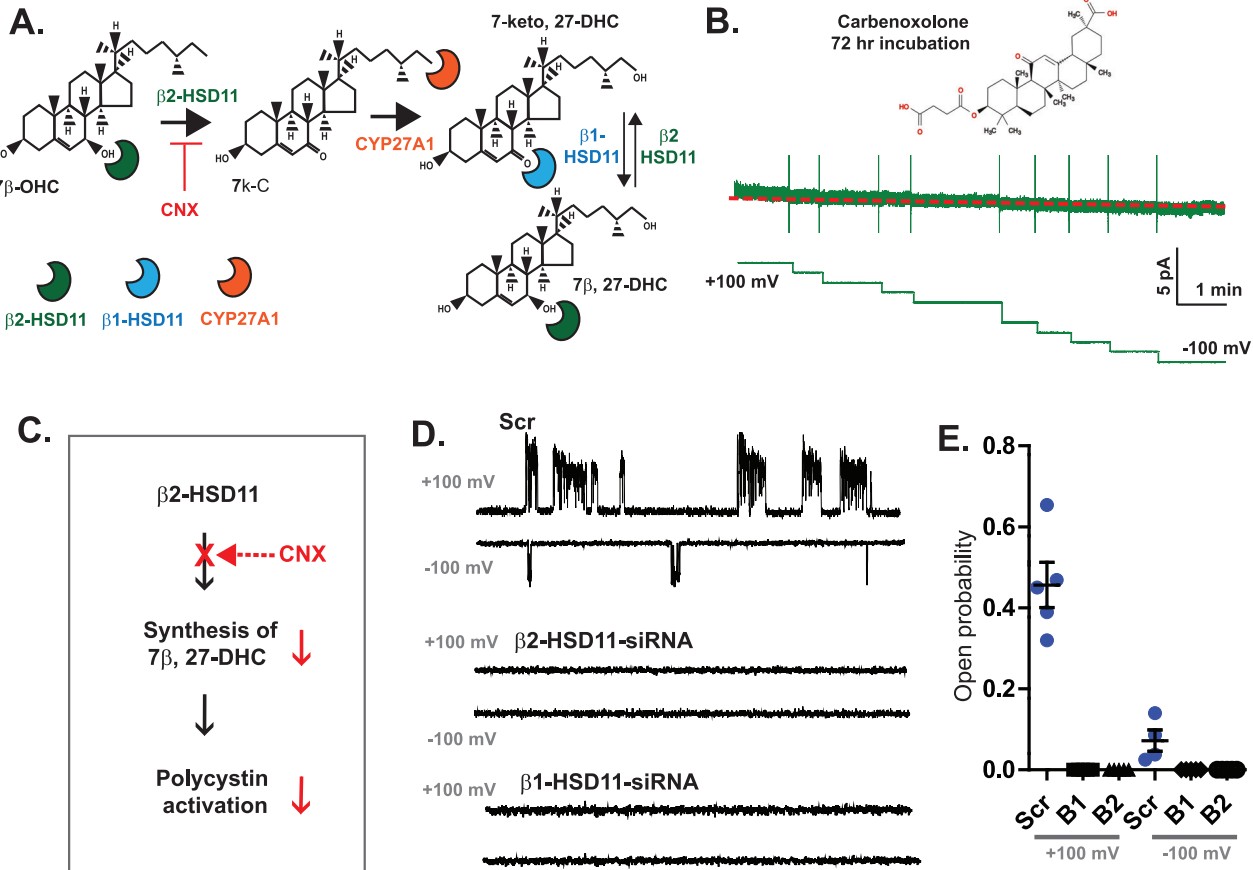

**Fig. 6 | 7β,27-DHC synthesis is critical for basal activity of ciliary polycystin channels. A** Schematic pathway of 7β,27-DHC synthesis. Schematic diagram illustrating key role of β1/2-hydroxysteroid dehydrogenase type 11 (β1/2-HSD11) enzymes for conversion of 7-keto and 27-HC to 7β,27-DHC. Carbenoloxone (CNX) inhibits enzymatic activity of β2-HSD11. **B** Excised ciliary single-channel recording of IMCD-3 cells after 72-h CNX incubation. Holding potentials were given from +100 mV to −100 mV. Red dotted line indicates closed state. **C** Model of CNX-dependent polycystin inhibition. **D** Ciliary single-channel recordings after β1/2-HSD11 knockdown. IMCD-3 cells were transfected with scrambled siRNA control (top), β2-HSD11-siRNA (middle), and β1-HSD11-siRNA (bottom). Knockdown efficiency was 95 ± 7% for β2-HSD11 and 90 ± 5% for β1-HSD11. Single channels with recorded in ciliary excised patches at +100 mV and −100 mV. **E** Quantification of open probability of ciliary recordings shown in **D**. Open events of single-channel recordings were counted at +100 mV and −100 mV. Scr (scrambled) $n = 5$, B1 (β1-HSD11) $n = 21$, B2 (β2-HSD11) $n = 23$. Error bars, mean ± SEM.

providing a tantalizing molecular explanation for the F604P GOF mutation (Supplementary Fig. 4A–C). We also compared the channel opening kinetics between PC-2 F604P and 7β,27-DHC-dependent WT activation by measuring the time constant (τ, τ) at depolarizing potential (+180 mV) (Supplementary Fig. 4D and E). Channel opening of the GOF mutant ($\tau = 0.06 \pm 0.00$ ms, $n = 4$) or sPC-1/PC-2 + 7β,27-DHC ($\tau = 0.05 \pm 0.01$ ms, $n = 4$) was comparable. Together, these findings demonstrate that 7β,27-DHC can further promote opening of ciliary polycystin channels at physiological membrane potentials and suggest that endogenous ciliary 7β,27-DHC concentrations are not saturating polycystin activation.

### Oxysterol-binding pocket is present in PC-2 but not in other polycystin members

Based on protein sequence alignments[58,59], E208 and R581 in PC-2 are not conserved in PC-2L1 and PC-2L2 (Supplementary Fig. 5A), suggesting that PC-2 is the only 7β,27-DHC−sensitive polycystin member. As expected, inside-out single-channel recordings in HEK293 cells transiently expressing PC-2-L1-EGFP did not show any potentiation with 5 μM 7β,27-DHC (Supplementary Fig. 5B). In addition, PC-2L1 did not result in significantly different conductance in outward ($G_{PC-2L1+7\beta,27-DHC}$: 208.0 ± 7.8 pS, $n = 6$, $G_{PC-2L1}$; 205.8 ± 10.8 pS, $n = 5$) and inward currents ($G_{PC-2L1+7\beta,27-DHC}$: 89.0 ± 22.0 pS, $n = 6$, $G_{PC-2L1}$;

107.0 ± 12.0 pS, $n = 5$) (Supplementary Fig. 5C, D). The absolute open probability of PC-2L1 was not affected by 7β,27-DHC (Supplementary Fig. 5E), suggesting that 7β,27-DHC is specific for PC-2.

### Synthesis of ciliary oxysterols by 11β-HSD enzymatic activity is essential for polycystin activation

Having established that 7β,27-DHC is a sufficient to activate the polycystin complex in both the plasma and ciliary membranes, we next asked whether 7β,27-DHC is required for polycystin activity in primary cilia. 11β-Hydroxysteroid dehydrogenase (11β-HSD) is a key enzyme to generate oxysterols, including those found in cilia[8,60]. Carbenoxolone (CNX), a component of licorice, is known to inhibit the activity of 11β-HSD[8] (see also Fig. 6A). A recent study demonstrated that CNX-dependent inhibition of 11β-HSD2 impairs activation of the hedgehog pathway[8]. As shown in Fig. 6B, incubation of IMCD-3 cells with CNX for 72 h completely abrogated endogenous polycystin channel activity (Fig. 6B), even during long recordings ($n = 20$). Next, we assessed the effects of genetic inhibition of 7β,27-DHC using small interfering RNA (siRNA) targeted against 11β-HSD1 and 11β-HSD2. Depletion of both enzymes in IMCD-3 cells resulted in a complete loss of ciliary polycystin currents compared with those observed in cells treated with scrambled siRNA (Fig. 6D, E), further emphasizing the notion that oxysterols are required for ciliary polycystin activity in IMCD-3 cells.

## Discussion

The present study identified a cilia-enriched oxysterol, 7β,27-DHC, as a critical cofactor for polycystin activation. This study was initiated in response to the puzzling observation that WT PC-2 remains silent in the plasma membrane when co-expressed with sPC-1, despite its efficient insertion into the plasma membrane[28]. This let us to speculate that cofactors found specifically in cilia may be required for channel activation. Our current model suggests that cilia-specific lipids confer spatial specificity for activating the polycystin complex in the ciliary membrane. While PC-2 has been reported in several subcellular locations, including the endoplasmic reticulum[61], organelle-specific cofactors may license channel activation only in the cilium. This concept is common in channel physiology: for instance, mucolipins (TRPML's), the closest TRP channel homologs to polycystins, require $PI(3,5)P_2$ for activation. $PI(3,5)P_2$ is enriched in the endosomal membrane, thus restricting TRPML activity to endosomes[62–64]. While $PI(3,5)P_2$ and $PI(4,5)P_2$ are well established second messengers required for TRP channel activation, less is known about ciliary enriched PI(4)P as an ion channel modulator. This opens the interesting possibility that PI(4)P regulates ciliary protein function[2,65].

Furthermore, Wang et al[57]. proposed a potential binding of phosphatidylinositol-4,5-bisphosphate ($PIP_2$) to the PC-2 subunit, suggesting a pivotal role for phosphoinositides in modulating the function of polycystins[57]. Our results point towards a potential competition among different functional classes of lipids to modulate polycystins. While we noted impaired 7β,27-DHC-dependent activation for non-ciliary PIPs, only PI(4)P completely abrogated polycystin currents. PI(4)P is generated within primary cilia by dephosphorylation of $PIP_2$ by cilia-localized $PIP_5$ phosphatase INPP5E, and abnormal ciliary PI(4)P levels cause a cystic phenotype[66]. Future studies will need to address the physiological concentrations of phosphoinositides that are required for 7β,27-DHC competition. Our current working model proposes that 7β,27 licenses polycystin activation to the primary cilium, while dynamic regulation by PI(4)P may finetune channel activity. Alternatively, 7β,27-DHC could also be dynamically regulated while PI(4)P levels set the overall threshold. Thus, potential biosensors are required to study the regulation of oxysterols in primary cilia. While other oxysterols structurally related to 7β,27-DHC failed to activate the polycystin complex, we cannot rule out that other cofactors may contribute to channel activation. Using molecular dynamics simulations, we predicted a potential binding site for 7β,27-DHC in PC-2. Interestingly, Wang et al. also predicted a location near E208 and R581 as a potential phospholipid or cholesterol-binding pocket in human PC-2[57]. E208 and R581 are not conserved among other PC-2 family proteins, including PC-2L1 and PC-2L2, suggesting that PC-2 is a unique lipid-binding protein. Indeed, a proteomics study identified PC-2 as the only sterol-binding protein among the TRP channel family[56].

However, the clinical correlation between oxysterols, polycystins, and ADPKD remains unclear. 11β-HSD2 is best described in the context of glucocorticoid synthesis and deficiency in 11β-HSD2 can cause inappropriate glucocorticoid activation of renal mineralocorticoid receptors (MRs)[67,68]. Clinically, deficiency in 11β-HSD2 in the kidney causes apparent mineralocorticoid excess (AME) syndrome, which is characterized by hypertension and hypokalemia[43,69–72]. Moudgil et al. reported renal cysts in an AME patient, showing that inhibited 11β-HSD2 enzymatic activity may lead to renal cyst formation[73]. Furthermore, treatment of newborn mice with high-dose glucocorticoids induced polycystic kidney disease[74], indicating that cystogenesis might associate with activation of 11β-HSD enzymes. However, knockout of CYP11B1 or CYP11B2, the two genes encoding 11β-HSD1 and 11β-HSD2 in mouse models do not result in kidney cysts[75,76]. The redundancy in 7β,27-DHC synthesizing enzymes is currently only poorly understood[44,45,77]. Given that the heterozygous *PKD2* mouse model does not induce renal cysts, we hypothesize that depletion of 11β-HSD expression may require the heterozygosity of *PKD2* to induce

renal cystogenesis, mimicking the third-hit model[78]. Thus, further clinical investigations are required to determine the precise mechanism of renal cyst development by inhibiting oxysterol synthesis.

Together the results of this study show that the cilia-specific oxysterol, 7β,27-DHC, can potentiate polycystin channel activity. We also show that 7β,27-DHC binding to PC-2 is enantioselective, highlighting the specificity of the interaction. Recent research demonstrated that re-expression of PC-2 can reverse renal cystogenesis, indicating that the activity of polycystins suppresses cilia-dependent cyst-activating (CDCA) signals[40]. Our study also suggests that the development of polycystin complex activators that specifically target the oxysterol-binding site is a promising therapeutic approach to compensate for ADPKD-causing loss of function mutations in PC-2.

## Methods

### Cell culture and transfection

Human embryonic kidney (HEK) 293 cells and mouse inner medullary collecting duct 3 (mIMCD-3) were obtained from ATCC, transfected with the indicated vectors, and used for whole-cell patch-clamp and single-channel recordings from excised cilia, respectively. HEK293 cells were cultured in Dulbecco's Modified Eagle Medium (DMEM; Gibco) supplemented with 10% fetal bovine serum (FBS) and 0.1% penicillin-streptomycin at 37 °C in a $CO_2$ incubator (Heraeus). mIMCD-3 cells were cultured in F-12K Nutrient Mixture (1X) Kaighn's Modification (Gibco) supplemented with 10% FBS and 0.1% penicillin-streptomycin. For cilia formation, mIMCD-3 cells were incubated in serum-free OPTI-MEM (GIBCO) for 48 h at 37 °C and 5% $CO_2$.

The sPC-1/2 construct was cloned into pTRE vectors, and HEK293 cells were transfected using Lipofectamine 2000 (Invitrogen), according to the manufacturer's instructions. sPC-1/2–transfected HEK293 cells were treated with 1 µM doxycycline 12 h before the experiment. For the patch-clamp experiment, the cells were seeded into the patch-clamp chamber. For the ciliary patch-clamp test, mIMCD-3 cells were transfected with ARL13B-GFP using Lipofectamine LTX (Invitrogen), according to the manufacturer's instructions.

### Immunocytochemistry and imaging

All procedures were performed at room temperature. Cells were fixed in 3.2% paraformaldehyde for 10 min, permeabilized in 0.2% Triton X −100 for 10 min, and blocked in blocking solution (BS) that contained 5% fetal calf serum (FCS), 2% bovine serum albumin (BSA), 0.2% fish gelatin, and 0.05% $NaN_3$ in PBS for 30 min. Cells were then incubated with primary antibody solution (BS plus primary antibody) for 1 h. Antibodies used: rat anti-HA (3F10, Roche), 1:1000; mouse anti-Arl13b (N295B/66, Antibodies Incorporated), 1000; rabbit anti-HA (C29F4, Cell Signaling), 1:1000; mouse anti PKD2 (D3, Santa Cruz Biotechnology), 1:200. Cells were washed two times with PBS before incubation with secondary antibody solution (BS plus fluorescent labeled secondary antibody (Thermo Fisher) and Hoechst at 1:1000 dilution) for 1 h. Cells were repeatedly washed with PBS and mounted onto glass coverslips for imaging.

To stain for surface HA-tag expression, live cells were incubated in rat anti-HA (3F10, Roche) primary antibody at a concentration of 1:100 in Leibovitz's L-15 medium for 20 min. Cells were washed once with L-15 medium, fixed and stained as described above.

Mounted cells were imaged on a Nikon Ti inverted fluorescence microscope with CSU-22 spinning disk confocal and EMCCD camera. Z-stacks were acquired to visualize the full z-dimension of each cilium. To compare trafficking between WT and mutant PC-2, imaging acquisition parameters were kept constant for all conditions. Ciliary and cell membrane surface intensities were calculated in ImageJ (NIH).

### siRNA transfection and quantification of RNA levels

Arl13b-enhanced green fluorescent protein (EGFP)–expressing IMCD-3 cells were transfected with 20 pM of siRNA (Dharmacon) and 7.5 µl

RNAiMAX (Life Technologies) in a 3.5-cm dish. Cells were seeded at 230,000 cells/plate. After 48 h of incubation for transfection, cells were serum-starved, and cilia recordings performed 48 h after serum starvation. For expression analysis, IMCD-3 cells in a 3.5-cm dish were washed once with PBS and lysed in 1 ml TRIzol (Zymo Research) for RNA extraction according to the manufacturer's instructions. RNA was reverse transcribed using the QuantiTect reverse transcription kit (Qiagen). Gene-specific primers were designed using Primerbank (http://pga.mgh.harvard.edu/primerbank/). Real-time PCR was performed, and the results analyzed on a BioRAD CFX284 according to the manufacturer's instructions. Primer sequences for *11β-HSD1* (up: cagaaatgctccagggaaagaa, dn: gcagtcaataccacatgggc); *11β-HSD2* (up: ggttgtgacactggttttggc, dn: agaacacggctgatgtcctct).

## Electrophysiological recordings

For a whole-cell patch-clamp experiments HEK293 cells with tetracycline-inducible, stably integrated expression vectors were used. The glass electrodes (Sutter Instruments, BF150-86-10) were prepared using a micropipette puller (Sutter Instruments, SU-P1000). Tips of glass electrodes were polished using a micro forge (Narishige, MF-830), resulting in a bath resistance range of 8–10 MΩ. mIMCD-3 cells were used for single-channel recordings from excised cilia using forged glass electrodes with bath resistance in the range of 26–30 MΩ.

Data were acquired with Multiclamp 200B amplifier (Molecular Device, Axon Instruments), filtered at 5 kHz. Data were digitized at 10 kHz using Digidata 1324 A (Molecular Device, Axon Instruments). The reference electrode was grounded using a 3 K KCl agar-bridge.

For the whole-cell patch-clamp recordings of HEK293 cells, both step pulse and ramp pulse protocols were used. The step pulse protocol consisted of voltage steps from −100 mV to +180 mV in 20-mV increments. The length of each step was 150 ms. The holding potential and tail pulse were given at 0 mV and −80 mV, respectively. The ramp pulse protocol was gradually applied from −100 mV to +180 mV during 500 ms with 0 mV holding potential.

The intracellular solution contained (in mM): 90 sodium methanesulfonate (NaMES), 10 NaCl, 10 HEPES, 5 EGTA, 2 $MgCl_2$ and 0.1 $CaCl_2$ (free $Ca^{2+}$ -100 nM, Maxchelator). Solution was adjusted to pH 7.4 using NaOH and 290 ± 5 mOsm/kg using D-mannitol. For N-methyl-D-glucamine (NMDG) test in Supplementary Fig. 2C and D, the intracellular solution contained (in mM): 70 NMDG, 5 EGTA, and 100 HEPES, adjusted to pH 7.4 using gluconic acid and 270 ± 5 mOsm/kg using D-mannitol.

The extracellular solution contained (in mM): 145 sodium gluconate (NaGlu), 5 KCl, 2 $CaCl_2$, 5 $MgCl_2$ and 10 HEPES. Solution was adjusted to pH 7.4 using NaOH and 290 ± 5 mOsm/kg using D-mannitol. Extracellular NMDG solution contained (in mM): 100 HEPES, 100 NMDG and 5 EGTA. It was adjusted to pH7.4 using gluconic acid and 290 ± 5 mOsm/kg using D-mannitol. The osmolarity of all solutions was measured using a vapor pressure osmometer (VAPRO, Wescor, Inc.).

## Data analysis

The whole-cell patch-clamp data and single-channel data from excised cilia were recorded using Clampex 10.0 and analyzed using Clampfit 10.6 (Molecular Devices).

For the whole-cell patch-clamp, the current (I; pA) and voltage (V; mV) relationship was assessed by plotting the maximum current amplitudes obtained at each step pulse divided by capacitance (pF). For the comparison of the current density, maximum current amplitudes at +180 mV divided by capacitance (pF) were plotted and compared in a bar graph using Prism 5.0.

For single-channel recording from excised cilia, the current amplitudes were plotted in a conventional histogram and fitted to the

Gaussian equation at each potential.

$$f(x) = \sum_{i=1}^{n} Ai \frac{e^{-(x-\mu i)^2}/2\sigma i^2}{\sigma i \sqrt{2\Pi}} + C \quad (1)$$

where n is the component, A is the amplitude, μ is the Gaussian mean, σ is the Gaussian standard deviation and C is the constant y offset for each i component. After fitting the Gaussian fitting, $\mu 1$ is subtracted from $\mu 2$, and value for each voltage potential is plotted to show the current (I; pA) and voltage (V; mV) relationship. To calculate conductance (G; pG), currents were plotted from −100 to +100 mV voltage potentials and fitted to linear equations.

To calculate the single-channel open probability, 10 s of the recording were selected at each voltage potential, and the open probability was calculated using the following equation:

$$Popen = \frac{To}{T} \quad (2)$$

where To is the total time that the channel presented in the open state and T is the total observation time. If a patch contained more than one of the same type of channel, $P_{open}$ was computed by:

$$Popen = \frac{To}{NT} \quad (3)$$

where N indicates the number of channels in the patch. The following equation was used to populate data.

$$To = \sum LTo \quad (4)$$

where L indicates the level of the channel opening. The absolute probability of the channel being open NPo was computed by:

$$Npo = \frac{To}{To + Tc} \quad (5)$$

where Tc indicates the total time in the closed state.

To measure the time constant (τ) of channel activation, the current amplitudes at +180 mV were plotted and fitted to the following exponential decay equation:

$$Y = Y_0 + Ae^{-\chi/\tau} \quad (6)$$

where $Y_O$ indicates the initial offset; $A$ is the amplitude of the oscillation; $\chi$ is the decay constant that determines the rate of decay; and $\tau$ is the time elapsed since the initial displacement.

## Computational methods

The homomeric PC-2 tetramer complex was simulated in the presence of 7β,27-DHC to sample free binding events. The protein structure from Protein Data Bank 5T4D[25] was embedded into the 1-palmitoyl-2-oleoyl-glycero-3-phosphocholine (POPC) lipid bilayer and neutralized in 150 mM NaCl, using the CharmmGUI webserver[79]. Eight 7β,27-DHC molecules were placed in solution above the upper leaflet and below the lower leaflet. The total system size was ~300 K atoms. Simulations were carried out using the ff14SB AMBER[80] parameter set for the protein, the Joung–Cheatham[81] parameters for the monovalent ions, LIPID17[82] for the lipids, and the general Amber force field (GAFF)[83] for 7β,27-DHC. The TIP3P model was used to simulate water. The 7β,27-DHC geometry was first optimized, and its electrostatic potential was calculated with Gaussian v9-E01[6] with the 6−31 G* basis set. Next, the restricted electrostatic potential (RESP) charges were fitted into the

electrostatic potential with antechamber, and the force field parameters, topology, and starting coordinates were generated with AmberTools. Simulations were carried out with the Particle Mesh Ewald Molecular Dynamics (PMEMD) engine on graphics processing units (GPUs). Minimization consisted of 10,000 steps, switching from the steepest descent algorithm to conjugated gradient after 5000 steps. The system was then gradually heated from 0 to 303.15 K over 100 ps, and harmonic restraints with a spring constant of 10 kcal/mol/$Å^2$ were applied to all heavy atoms except water oxygen. After reaching 100 K, we switched from NVT (constant temperature, constant volume) to NPT (constant temperature, constant pressure). After 100 ps, force constants of 10, 5, and 2.5 kcal/mol/$Å^2$ were applied to the substrates and nearby residues (within 5 Å), protein, and lipid headgroups, respectively. Restraints were gently removed over the next ~10 ns, and the homomeric system was simulated for ~2 µs. Pressure (1 bar) was maintained using a semi-isotropic pressure tensor and the Monte Carlo barostat. Temperature was maintained with the Langevin thermostat with a friction coefficient of $1\,ps^{-1}$. The SHAKE algorithm was used with a 2-fs time step. A non-bonded cutoff of 10 Å was used, and electrostatics were calculated using the particle mesh Ewald method.

## Statistical analysis

Data were analyzed using Clampfit 10.6 (Axon Instruments, Molecular Devices), OriginPro (Origin Lab) and Prism (GraphPad). The mean values are presented with the standard error of the mean (SEM), and the number of independent experiments or observations (N) is specified for electrophysiology (number of cells), animal studies, and quantified imaging. Statistical significance was determined using paired or unpaired two-tailed Student's $t$-test, or one- or two-way analysis of variance (ANOVA), with a significance threshold of $P < 0.05$. The sample sizes were determined based on the number of independent experiments required for statistical significance and technical feasibility.

## Reporting summary

Further information on research design is available in the Nature Portfolio Reporting Summary linked to this article.

## Data availability

The data that support this study are available from the corresponding authors upon request. The published structural model used for Fig. 3C, D is available at 5T4D. Molecular dynamics trajectories have been deposited in Zenodo [https://zenodo.org/records/12511678]. Source data are provided with this paper.

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

## Acknowledgements

This work was supported by National Institute of Health Grant R01DK127277 (MD and EC), a National Research Foundation of Korea (NTF) grant funded by the Korean government (MSIT) (No. 2019R1A6A3A03033302) (KH), National Institute of Health Grant 1K99DK131361-01A1 (KH), and the PKD Foundation (A137178) (KH). A histologic examination was performed by Marcella Foti at the Gladstone Institute. A portion of this work was performed under the auspices of the U.S. Department of Energy by Lawrence Livermore National Laboratory under Contract DE-AC52-07NA27344 (PB).

## Author contributions

KH and MD designed the project. KH performed the electrophysiological experiments. MD generated the cell lines and AP maintained the cell lines. MD performed siRNA experiment and qRT-PCR analysis. NM performed the imaging analysis. PB simulated the computational molecular dynamics. DRR shared the oxysterols. JFR, GL, EC, and MD provided supervision. The manuscript was drafted by KH and MD. All authors critically reviewed the text.

## Competing interests

M.D. and J.R. are cofounders of a 5AM Venture-backed Newco. The remaining authors declare no competing interests.
