## [Peer Review File · Nature Communications]

Cilia-enriched oxysterol 7 β ,27-DHC is required for polycystin ion channel activationREVIEWER COMMENTS

Reviewer #1 (Remarks to the Author):

In the manuscript "Cilia associated oxysterol 7beta,27-DHC is required for polycystin ion channel activation" the authors, Ha et al, describe a role for 7beta,27-DHC in the activation of the polycystin channel complex. The work has the potential to be very impactful in the ADPKD field, a field that has struggled to understand the function and localization of the polycystin complex and how it controls tubular and cellular shape. The experiments presented are straight forward and well executed but there are a few points that need further exploration to complete this exciting story.

1. A previous group (Wang et al 2020) has shown that PC2 binds multiple species of phospholipids and modeled the area of interaction to the S3, S4, S5 intersection/ pocket of PC2. I appreciate the experiments the authors have done in the supplement to evaluate if some of the PIPs activate the sPC1/PC2 channel complex. But the predicted PC2 / 7beta,27-DHC interaction area is in the same pocket as the PIP binding area. So how do the phospholipids compete with 7beta,27-DHC for binding to PC2? Given the PIPs are membrane specific (cilia, PM) if there are differences in the competition for binding within the PIPs this may strengthen the authors' arguments that the polycystin channel complex is primarily active in the cilium (ie PIP2 prevents 7beta,27-DHC activation but PIP does not). The authors should repeat the experiments of supplemental figure 1 with both the different phospholipids and with 7beta,27-DHC to assess how relative concentrations of PIPs and differences in PIP effects affect the 7beta,27-DHC activation of PC1/PC2.

2. The second main point is there is insufficient evidence to support the authors' assertions that the work presented demonstrates that 7beta,27-DHC limits "polycystin-mediated signaling to this unique [ciliary] cellular compartment". This is based on previous work like the excellent Raleigh et al 2018 that convincingly showed 7beta,27-DHC was enriched in the cilia of sea urchin embryos. However they also showed that 7beta,27-DHC was in the de-ciliated embryos and therefore not exclusive to the cilium. The authors should consider removing the conclusion that the activation of the PC1/PC2 complex by 7beta,27-DHC is cilia specific. If they wish to claim this then they will need to do considerable further experiments: Show association of 7beta,27-DHC in cilia of renal epithelial cells of at least proximal and collecting duct segments, and show that there is effectively no 7beta,27-DHC in the other membranes where PC1/PC2 activation has been suggested / demonstrated (PM and ER).

Here the authors could simply conclude that 7beta,27-DHC is cilia associated / enriched and supports the activation of the PC1/PC2 in this very important cellular compartment.

In addition to the major comments above there are some other issues that need clarification:

Figure 1

a) PC1-deltaNT/PC2. In the previous Ha et al 2020 publication, the authors showed the PC1-deltaNT/PC2 complex made it to the surface but could not be activated. This is the CTF version of PC1. If it makes it to the membrane can the 7beta,27-DHC activate the channel complex? This might provide some understanding of how the cleaved NTF and 7beta,27-DHC interact to control channel activation.

Figure 2

a) How many 7beta,27-DHC molecules are predicted to bind to PC2? Why is a 10X increase in 7beta,27-DHC necessary to increase open probability for 2 channels or so per patch? Can the number of 7beta,27-DHC molecules binding per PC2 be calculated, or how long they stay bound etc.?
b) Figure 2C, there is no dose response – that would require significant increases (or decreases) at each discrete 7beta,27-DHC concentration as compared to the previous concentration.

Figure 3

a) How does 7beta,27-DHC interaction alter the PC2 permeation pore? Can it be modeled?

Figure 4

a) G: Are these channel recording done with or without $7\beta,27\text{-DHC}$? Please clarify in legend.

Figure 6

a) Knocking down HSD11B1/2 in IMCD cells is good, but expression of these genes in HEK cells and showing sPC1/PC2 activation in plasma membrane from cell attached recording is a critical experiment: it may either support, or not, the claim this is a ciliary only activation process.

Reviewer #2 (Remarks to the Author):

Mutations of TRPP2(PC-2) and PKD1(PC-1) cause Autosomal Dominant Polycystic Kidney Disease (ADPKD), a common genetic disease in humans. PC-1 and PC-2 are known to form a heteromeric non-selective cation channel (PC-1/2) in the primary cilia of kidney epithelial cells, but how PC-1/PC-2 is activated by endogenous cellular cue(s) is not clear. In a previous study, the authors showed that an N-terminal peptide segment of PC-1 serves as an intrinsic agonist of the PC-1/2 channel. In the current study, the authors found that $7\beta, 27\text{-DHC}$, a cilia-specific oxysterol, is not only required for the channel activity of PC-1/2 in the primary cilia, but also sufficient to activate PC-1/2 heterologously expressed at the plasma membrane of HEK293 cells. Overall, the agonist effects of $7\beta, 27\text{-DHC}$ was clearly demonstrated using both whole-cell recordings and excised patch recordings from ciliary membranes. Given the clinical and physiological significance of the PC-1/2 channel, these findings are of importance and interest to both PKD and ion channel fields. The manuscript could be improved if following concerns can be addressed.

1. Can you explain why some whole-cell electrophysiological traces, e.g, Fig. 1D & Fig. 1I, appeared to be quite noisy?
2. What was the rationale to use a pipette solution ($\sim 100 \text{ mM Na}^+$) with low ionic strength? More importantly, given that the extracellular/bath solution contained more than 145 mM Na^+ , I expected a positive reversal potential ($> +20 \text{ mV}$) for the PC-1/2 currents. However, in Fig.1 E, the reversal potential is close to 0 mV or negative. It is also not clear how the endogenous TRPM7-like currents were blocked in the recording conditions.
3. Given that the PC-1/2 current is outwardly-rectifying, the NMDG⁺ control experiment in Fig. 1I should be performed using NMDG⁺ in the pipette solution. Can you explain why bath perfusion of a NMDG⁺ solution caused a reduction in the outward (presumably due to the efflux of intracellular Na^+ in the pipette solution)? On the other hand, the NMDG⁺ control experiment can be easily performed in the inside-out recordings via bath perfusion.
4. What was the inset panel in Fig. 1D---PC-2?

Answers to the reviewers' comments

Here, we thank the reviewers for their positive comments and insightful suggestions for our manuscript. We addressed each comment to the best of our knowledge, with our responses highlighted in blue.

Reviewer #1

In the manuscript “Cilia associated oxysterol 7 β ,27-DHC is required for polycystin ion channel activation” the authors, Ha *et al*, describe a role for 7 β ,27-DHC in the activation of the polycystin channel complex. The work has the potential to be very impactful in the ADPKD field, a field that has struggled to understand the function and localization of the polycystin complex and how it controls tubular and cellular shape. The experiments presented are straight forward and well executed but there are a few points that need further exploration to complete this exciting story.

1. A previous group (Wang *et al* 2020) has shown that PC2 binds multiple species of phospholipids and modeled the area of interaction to the S3, S4, S5 intersection/ pocket of PC2. I appreciate the experiments the authors have done in the supplement to evaluate if some of the PIPs activate the sPC1/PC2 channel complex. But the predicted PC2 / 7 β ,27-DHC interaction area is in the same pocket as the PIP binding area. So how do the phospholipids compete with 7 β ,27-DHC for binding to PC2? Given the PIPs are membrane specific (cilia, PM) if there are differences in the competition for binding within the PIPs this may strengthen the authors arguments that polycystin channel complex is primarily active in the cilium (ie PIP2 prevents 7 β ,27-DHC activation but PIP does not). The authors should repeat the experiments of supplemental figure 1 with both the different phospholipids and with 7 β ,27-DHC to assess how relative concentrations of PIPs and differences in PIPs effects the 7 β ,27-DHC activation of PC1/PC2.

Answer | Following the reviewer's suggestion, we tested in whole-cell recordings whether phosphatidylinositols might compete with or modulate 7 β ,27-DHC-dependent activation of the polycystin complex. Specifically we tested 5 μ M of PI(3,4)P₂, PI(4,5)P₂, and PIP₄ applied along with 7 β ,27-DHC to the intracellular solution. While PI(3,4)P₂ and PI(4,5)P₂ only reduced current density of 7 β ,27-DHC evoked polycystin currents, addition of PIP₄ completely counteracted 7 β ,27-DHC-dependent channel activation, suggesting a potent competition between PIP₄ and 7 β ,27-DHC (New Supplementary Figure 3A and B). While these findings were initially surprising to us (we had expected results similar to what reviewer #1 implied, such as polycystin inhibition by PI(4,5)P₂ to ensure activation only within the ciliary compartment). However, Garcia-Gonzalo *et al.* (2015) reported that PIP₄ is dynamically regulated within primary cilia by cilia-localized PIP₅ phosphatase INPP5E and that abnormal ciliary PIP₄ levels cause a cystic phenotype. Based on these findings, we now hypothesize that ciliary 7 β ,27-DHC licenses polycystin activation, while changes in PIP₄ levels may acutely regulate polycystin channel activity. Thanks to the reviewer, we will investigate this exciting new avenue of PIP₄-dependent polycystin regulation in future studies.

2. The second main point is there is insufficient evidence to support the authors assertions that the work presented demonstrates that 7 β ,27-DHC limits “polycystin-mediated signaling to this unique [ciliary] cellular compartment”. This is based on previous work like the excellent Raleigh *et al* 2018 that convincingly showed 7 β ,27-DHC was enriched in the cilia

of sea urchin embryos. However, they also showed that $7\beta,27\text{-DHC}$ was in the de-ciliated embryos and therefore not exclusive to the cilium. The authors should consider removing conclusion that the activation of the PC1/PC2 complex by $7\beta,27\text{-DHC}$ is cilia specific. If they wish to claim this then they will need to do considerable further experiments: Show association of $7\beta,27\text{-DHC}$ in cilia of renal epithelial cells of at least proximal and collecting duct segments, and show that there is effectively no $7\beta,27\text{-DHC}$ in the other membranes where PC1/PC2 activation has been suggested / demonstrated (PM and ER). Here the authors could simply conclude that $7\beta,27\text{-DHC}$ is cilia associated / enriched and supports the activation of the PC1/PC2 in this very important cellular compartment.

Answer | We thank the reviewer for these insightful thoughts. We agree fully with the reviewer and made the necessary adjustments. Following the reviewer's suggestion, we revised the title to "**Cilia-Enriched Oxysterol $7\beta,27\text{-DHC}$ is required for Polycystin Ion Channel Activation.**" Additionally, we have amended text and conclusion to propose that " $7\beta,27\text{-DHC}$ plays a crucial role in supporting the activation of the PC1/PC2 complex within this pivotal cellular compartment."

In addition to the major comments above there are some other issues that need clarification:
Figure 1

a) PC1-deltaNTF/PC2. In the previous Ha *et al* 2020 publication, the authors showed the PC1-deltaNTF/PC2 complex made it to the surface but could not be activated. This is the CTF version of PC1. If it makes it to the membrane can the $7\beta,27\text{-DHC}$ activate the channel complex? This might provide some understanding of how the cleaved NTF and $7\beta,27\text{-DHC}$ interact to control channel activation.

Answer | Following the reviewer's recommendation, we now include whole-cell patch clamp recordings for $\Delta\text{NTF PC-1}$ in the presence of $5\mu\text{M}$ intracellular $7\beta,27\text{-DHC}$. As illustrated in Supplementary Figure 1A and B, the $\Delta\text{NTF PC-1}$ failed to elicit channel activation upon intracellular treatment with $7\beta,27\text{-DHC}$. This experiment reaffirms the critical role of PC-1 NTF in polycystin activation.

Figure 2

a) How many $7\beta,27\text{-DHC}$ molecules are predicted to bind to PC2? Why is a 10X increase in $7\beta,27\text{-DHC}$ necessary to increased open probability for 2 channels or so per patch? Can the number of $7\beta,27\text{-DHC}$ molecules binding per PC2 be calculated, or how long the stay bound etc.?

Answer | We computed one lipid-binding event per subunit during the 2 microsecond simulation. As shown in Figure 2B, we conducted inside-out patch clamp experiments to determine potency of various $7\beta,27\text{-DHC}$ concentrations ($1\mu\text{M}$ - $50\mu\text{M}$). Although we showed an example where two channels opened during the inside-out single-channel recordings at $50\mu\text{M}$ of $7\beta,27\text{-DHC}$, the absolute open probability was calculated based on single channel openings. As we describe in the methods section, the number of channel openings is independent of absolute open probability.

b) Figure 2C, there is no dose response – that would require significant increases (or decreases) at each discrete 7 β ,27-DHC concentration as compared to the previous concentration.

Answer | We agree with the reviewer's comment regarding Figure 2C. We have revised the title to "High concentration 7 β ,27-DHC treatment increases the open probability for inward currents," instead of "Dose Response."

Figure 3

a) How does 7 β ,27-DHC interaction alter the PC2 permeation pore? Can it be modeled?

Answer | At present, there is no polycystin structure available in the open conformation, which we could use as a reference point for simulation. Therefore, we are unable to predict the arrangement of the pore.

Figure 4

a) G: Are these channel recordings done with or without 7 β ,27-DHC? Please clarify in legend.

Answer | All recordings in Fig. 4G were conducted without exogenous application of 7 β ,27-DHC and instead relied on the presence of endogenous oxysterol. The main conclusion of this figure is that both potential binding mutations, R581A and E208A, remain silent in ciliary recordings. We have added this clarification in the figure legend (Figure 4G), as recommended by the reviewer.

Figure 6

a) Knocking down HSD11B1/2 in IMCD cells is good, but expression of these genes in HEK cells and showing sPC1/PC2 activation in plasma membrane from cell attached recording is a critical experiment: it may either support, or not, the claim this is a ciliary only activation process.

Answer | Although it is widely believed that HSD11B1/2 are the rate limiting enzyme in the synthesis of oxysterols, CYP27 (encodes for sterol 27-hydroxylase) is also required to hydroxylate oxysterol position 27 to synthesize 7 β ,27-DHC. Further, it is currently not well understood which additional enzymes are ultimately required to generate enantiospecific oxysterol isoforms. Thus we have only limited possibilities to increase isoform-specific oxysterol concentrations using overexpression. For instance, overexpression of HSD11B1 in HEK293 cells results in a 3.3-fold increase in cortisol production, another important metabolite of HSD11B1/2². It is unknown whether HEK293 cells also increase 7 β ,27 levels after HSD11B1/2 overexpression. It remains a major bottleneck in studying oxysterols that we are lacking biosensors for specific oxysterol isoforms and thus have to rely on mass spectrometry and chromatography to accurately determine 7 β ,27-DHC levels. We thus feel that knock down of HSD11B in ciliated IMCD3 cells is the most direct approach we currently have. We added the caveats raised by the reviewer in the discussion.

Reviewer #2

Reviewer #2 (Remarks to the Author):

Mutations of TRPP2(PC-2) and PKD1(PC-1) cause Autosomal Dominant Polycystic Kidney Disease (ADPKD), a common genetic disease in humans. PC-1 and PC-2 are known to form a heteromeric non-selective cation channel (PC-1/ 2) in the primary cilia of kidney epithelial cells, but how PC-1/PC-2 is activated by endogenous cellular cue(s) is not clear. In a previous study, the authors showed that an N-terminal peptide segment of PC-1 serves as an intrinsic agonist of the PC-1/2 channel. In the current study, the authors found that 7b, 27-DHC, a cilia-specific oxysterol, is not only required for the channel activity of PC-1/2 in the primary cilia, but also sufficient to activate PC-1/2 heterologously expressed at the plasma membrane of HEK293 cells. Overall, the agonist effects of 7b, 27-DHC was clearly demonstrated using both whole-cell recordings and excised patch recordings from ciliary membranes. Given the clinical and physiological significance of the PC-1/2 channel, these findings are of importance and interest to both PKD and ion channel fields. The manuscript could be improved if following concerns can be addressed.

1. Can you explain why some whole-cell electrophysiological traces, e.g, Fig. 1D & Fig. 1I, appeared to be quite noisy?

Answer | Thank you very much for pointing this out! We addressed a technical issue with Adobe Illustrator that was causing the image to display noisy recording traces. We updated the figure in the revised version.

2. What was the rationale to use a pipette solution (~ 100 mM Na⁺) with low ionic strength? More importantly, given that the extracellular/bath solution contained more than 145 mM Na⁺, I expected a positive reversal potential (> + 20 mV) for the PC-1/2 currents. However, in Fig.1 E, the reversal potential is close to 0 mV or negative. It is also not clear how the endogenous TRPM7-like currents were blocked in the recording conditions:

Answers | As we describe in the methods section we used 145 mM Na-gluconate (NaGlu) in the pipette to mitigate contamination by endogenous calcium activated chloride channel (CaCC) in HEK293 cells. Regarding the reversal potential, Liu *et al.* 2018 and Kleene *et al.* reported that the polycystin complex is a non selective cation channel with strong preference for K⁺.^{3, 4} Given that both the intracellular and extracellular solutions contain all major cations, we assume that the reversal potential is mostly determined by K, thus resulting in a reversal potential close to zero.

To inhibit TRPM7-like currents, we included 5 mM MgCl₂ and 2 mM MgCl₂ in the extracellular and intracellular solutions, respectively. As the reviewer might be aware this is common practice in the TRP-channel field.

3. Given that the PC-1/2 current is outwardly-rectifying, the NMDG⁺ control experiment in Fig. 1I should be performed using NMDG⁺ in the pipette solution. Can you explain why bath perfusion of a NMDG⁺ solution caused a reduction in the outward (presumably due to the efflux of intracellular Na⁺ in the pipette solution)? On the other hand, the NMDG⁺ control experiment can be easily performed in the inside-out recordings via bath perfusion.

Answer | We appreciate the reviewer for pointing this out! To eliminate the possibility of the observed decrease in inward and outward currents due to an increase in membrane

resistance during recording, we now present a magnified view of the I-V curve in Figure 11. The shift in reversal potential (E_v) upon transitioning from NaGlu (-10.02 ± 6.4 mV) to NMDG extracellular solution (69.12 ± 5.4 mV) unequivocally validates that the altered currents are indeed a consequence of this solution change.

While Liu et al. (2018) demonstrated a reduction in inward currents but not outward currents with extracellular NMDG in endogenous ciliary ion channel recordings³, Shen et al.(2016) and Liu et al.(2023) reported that a PKD2-chimera or a gain of function mutation in PKD2 (PKD2-F604P) both show decreased outward and inward currents under similar conditions⁵.⁶ Based on these findings, we hypothesize that NMDG may partially block permeation. NMDG.

References.

1. Garcia-Gonzalo, F.R. *et al.* Phosphoinositides Regulate Ciliary Protein Trafficking to Modulate Hedgehog Signaling. *Dev Cell* **34**, 400-409 (2015).
2. Kragl, A. *et al.* Effects of HSD11B1 knockout and overexpression on local cortisol production and differentiation of mesenchymal stem cells. *Front Bioeng Biotechnol* **10**, 953034 (2022).
3. Liu, X. *et al.* Polycystin-2 is an essential ion channel subunit in the primary cilium of the renal collecting duct epithelium. *Elife* **7** (2018).
4. Kleene, S.J. & Kleene, N.K. The native TRPP2-dependent channel of murine renal primary cilia. *Am J Physiol Renal Physiol* **312**, F96-F108 (2017).
5. Shen, P.S. *et al.* The Structure of the Polycystic Kidney Disease Channel PKD2 in Lipid Nanodiscs. *Cell* **167**, 763-773 e711 (2016).
6. Liu, X. *et al.* Regulation of PKD2 channel function by TACAN. *J Physiol* **601**, 83-98 (2023).

REVIEWERS' COMMENTS

Reviewer #1 (Remarks to the Author):

In the manuscript "cilia enriched oxysterol 7beta,27-DHC is required for polycystin ion channel activation" Ha et al present compelling evidence that the oxysterol is a native activator of the polycystin channel complex in cilia. Their revisions have addressed my previous concerns. This work is an important step toward understanding the biological function of the polycystin in cilia.

Reviewer #2 (Remarks to the Author):

The authors have addressed my other concerns in the revision, but regarding comment #2, I was confused by the explanations in the rebuttal letter. You described in the Methods "The intracellular solution contained (in mM): 90 sodium methanesulfonate (NaMES), 10 NaCl, 10 HEPES, 5 EGTA, 2 MgCl₂ and 0.1 free calcium. This solution 570 was adjusted to pH 7.4 using NaOH and 290±5 mOsm/kg using D-mannitol". Hence, in your recording conditions (145 mM vs. 100 mM for Na⁺, and 5 mM vs. 0 mM for K⁺), both E(Na) and E(K) are positive values! In addition, with 5 mM EGTA, and 0.1 mM (added, not free Ca²⁺), intracellular [Ca²⁺] is less than 100 nM. Why would CaCCs (activated by micromolar intracellular Ca²⁺) contaminate your PC-1/2 currents? Also, you need to indicate that your extracellular solutions contained 5 mM Mg²⁺, not 1 mM, in order to block TRPM7-like endogenous currents. For the NMDG test in Fig. 1H, you only changed your extracellular solutions, right?

We thank the reviewer for insightful comments. Our responses highlighted in blue.

Reviewer #2 comment:

The authors have addressed my other concerns in the revision, but regarding comment #2, I was confused by the explanations in the rebuttal letter. You described in the Methods "The intracellular solution contained (in mM): 90 sodium methanesulfonate (NaMES), 10 NaCl, 10 HEPES, 5 EGTA, 2 MgCl₂ and 0.1 free calcium. This solution 570 was adjusted to pH 7.4 using NaOH and 290±5 mOsm/kg using D-mannitol". Hence, in your recording conditions (145 mM vs. 100 mM for Na⁺, and 5 mM vs. 0 mM for K⁺), both E(Na) and E(K) are positive values! In addition, with 5 mM EGTA, and 0.1 mM (added, not free Ca²⁺), intracellular [Ca²⁺] is less than 100 nM. Why would CaCCs (activated by micromolar intracellular Ca²⁺) contaminate your PC-1/2 currents? Also, you need to indicate that your extracellular solutions contained 5 mM Mg²⁺, not 1 mM, in order to block TRPM7-like endogenous currents. For the NMDG test in Fig. 1H, you only changed your extracellular solutions, right?

Answer: 1) We agree with the reviewers' concern regarding the polycystin reversal potential based on our initially used ion composition of recording solutions. As the reviewer points out, we should expect a slightly positive reversal potential (~+5-10mV). However, we observe a slightly negative reversal potential of -10mV. We assume that this slight discrepancy is due to junction potentials. To address the reviewers' concern we performed additional experiments and removed K⁺ from our solution and only measured Na⁺-evoked currents. Please see the table below for the composition of our bath solution and pipette solution. We also changed the typo in our methods section. The reviewer is correct, we had used 5mM Mg²⁺ all along in the extracellular solution to block TRPM7 currents.

Extracellular Solution				Intracellular Solution			
Substance	manufacturer (catalog number)	M.W (g/mol)	Conc (mM)	Substance	manufacturer (catalog number)	M.W (g/mol)	Conc (mM)
HEPES	Fisher Scientific (146601)	238.3	10	HEPES	Fisher Scientific (146601)	238.3	10
MgCl ₂	Sigma-Aldrich (M0250-500G)	203.3	5	NaMES	Acros Organics	118.09	100
Na-Gluconate	Sigma-Aldrich (A2054-1Kg)	218.14	145	EGTA	E0396-100g	380.35	5
CaCl ₂ dihydrate	Fisher Scientific (C76-500)	147.01	2	MgCl ₂	Sigma-Aldrich (M0250-500G)	203.3	5
				CaCl ₂ dihydrate	Fisher Scientific (C76-500)	147.01	1.94

Table 1. The composition of bath solution and pipette solution

Using these sodium-based of solutions, we measured the polycystin complex with 5 μM 7β,27-DHC in the pipette solution. As shown in Figure 1, we observed a right-shifted positive reversal potential as predicted by Nernst Equation.

We also corrected the typo in the methods section, indeed it was 0.1mM added Ca, not free Ca

Figure 1. The right shifted reversal potential with Na^+ solution.

2) In addition, with 5 mM EGTA, and 0.1 mM (added, not free Ca^{2+}), intracellular $[\text{Ca}^{2+}]$ is less than 100 nM. Why would CaCCs (activated by micromolar intracellular Ca^{2+}) contaminate your PC-1/2 currents?

Although there is only a slim chance that calcium activated chloride channel (CaCC) contaminate our polycystin recordings (as pointed out by the reviewer), we wanted to eliminate all potential chloride-based contaminations. Thus, we utilized Na-Gluconate and Na-Methanesulfonate in our solutions to eliminate Cl^- , except the Cl^- required for the reference electrode.

3) For the NMDG test in Fig. 1H, you only changed your extracellular solutions, right?

Yes, this is correct. We only changed composition of the extracellular solution for NMDG test. We also provide a detailed description of NMDG intracellular solution in the supplementary figure.